# Multiple myeloma: Combination therapy of BET proteolysis targeting chimeric molecule with CDK9 inhibitor

**Su-Lin Lim**[1☯]*, **Liang Xu**[2☯], **Bing-Chen Han**[1], **Pavithra Shyamsunder**[2], **Wee-Joo Chng**[2], **H. Phillip Koeffler**[1,2]

**1** Cedars Sinai Medical Center, Los Angeles, California, United States of America, **2** Cancer Science Institute of Singapore, National University of Singapore, Singapore, Singapore

☯ These authors contributed equally to this work.

* sulin_lim_86@hotmail.com

**Data Availability Statement:** All relevant data are within the paper and its Supporting Information files.

## Abstract

Cyclin Dependent Kinase 9 (CDK9) associates with Bromodomain and Extra-Terminal Domain (BET) proteins to promote transcriptional elongation by phosphorylation of serine 2 of RNAP II C-terminal domain. We examined the therapeutic potential of selective CDK9 inhibitors (AZD 4573 and MC180295) against human multiple myeloma cells *in vitro*. Short-hairpin RNA silencing of CDK9 in Multiple Myeloma (MM) cell lines reduced cell viability compared to control cells showing the dependency of MM cells on CDK9. In order to explore synergy with the CDK9 inhibitor, proteolysis targeting chimeric molecule (PROTAC) ARV 825 was added. This latter drug causes ubiquitination of BET proteins resulting in their rapid and efficient degradation. Combination treatment of MM cells with ARV 825 and AZD 4573 markedly reduced their protein expression of BRD 2, BRD 4, MYC and phosphorylated RNA pol II as compared to each single agent alone. Combination treatment synergistically inhibited multiple myeloma cells both *in vitro* and *in vivo* with insignificant weight loss. The combination also resulted in marked increase of apoptotic cells at low dose compared to single agent alone. Taken together, our studies show for the first time that the combination of a BET PROTAC (ARV 825) plus AZD 4573 (CDK9 inhibitor) is effective against MM cells.

## Introduction

Multiple myeloma (MM) is a clonal plasma cell malignancy. It is the second most common hematologic malignancy in United States [1]. Despite advances in treatment such as proteasome inhibitors and immunomodulatory drugs, the disease remains incurable. Bromodomain and Extra-Terminal Domain (BET) family is composed of BRD-2, -3, -4 and -T. They facilitates transcriptional activation by RNA polymerase II (RNAP II) [2]. ARV 825 (Arvinas, Inc) is a hetero-bifunctional molecule composed of a Bromodomain binding moiety (OTX 015) joined to pomalidomide. Pomalidomide binds to an intracellular E3 ubiquitin ligase, cereblon (CRBN); OTX 015 brings the complex to the BET molecules. This variety of inhibitor is called PROTAC (Proteolysis Targeting Chimeric molecules) which in this case causes ubiquitination

**Funding:** The sources of funding for this study are as follows: 1. Aaron Eshman and Melmed family, Morgan Stanley Inc. Recipient: H.P.K. 2. Leukemia and Lymphoma Society (LLS) Grant ID: 9002-16. Recipient: H.P.K. 3. Department of Defense (DoD) Award W81XWH-17-1-0093. Recipient: H.P.K. 4. UCLA Clinical and Translational Institute (CTSI) voucher V159. Recipient: H.P.K. 5. National Research Foundation Singapore under Singapore Translational Research (STaR). Investigator Award (NMRC/STaR/0021/2014) and administered by the Singapore Ministry of Health's National Medical Research Council (NMRC), the NMRC Centre. Grant awarded to National University Cancer Institute of Singapore, the National Research Foundation Singapore and the Singapore Ministry of Education under its Research Centers of Excellence initiatives. Recipient: H.P.K. 6. RNA Biology Center at the Cancer Science Institute of Singapore, NUS, Singapore Ministry of Education's Tier 3 grants, grant number MOE2014-T3-1-006. Recipient: H.P.K. The funders had no role in study design, data collection and analysis, decision to publish, or preparation of the manuscript.

**Competing interests:** HPK received funding from Morgan Stanley Inc and the Department of Defense (DoD). There are no patents, products in development or marketed products to declare. This does not alter our adherence to PLOS ONE policies on sharing data and materials.

of BET proteins resulting in rapid and efficient degradation of these proteins [3]. BET PRO-TAC ARV 825 inhibits the proliferation of MM cells both in vitro and in vivo [4,5].

Cyclin Dependent Kinase 9 (CDK9) is the kinase subunit of the positive transcription elongation factor b (P-TEFb) that associates with BET proteins which promotes transcriptional elongation by phosphorylation of serine 2 of RNAPII C-terminal domain (CTD) [6]. CDK9 has a major 42kDa and a minor 55kDa isoform. The 55kDa isoform is at an upstream transcriptional start site of the 42 kDa protein. Both are expressed in human cancer cell lines and in normal tissues [7]. The 42 kDa isoform is localized diffusely in the nucleoplasm, whereas the 55 kDa accumulates in the nucleolus [8]. CDK9 has been shown to play an important role in controlling global transcription, including expression of genes regulated by super-enhancers, such as MYC, MCL-1 and cyclin D1 [9]. MCL-1 and MYC are critical for proliferation of MM cells, often causing resistance to drugs and producing relapse in these patients [10,11]. Therefore, CDK9 may represent a druggable target in myeloma having dysregulated MYC expression [12,13].

Inhibition of both CDK9 and BRD 4 has been reported synergistically to induce growth arrest and apoptosis of cancer cells including MM [14,15]. Previously studied CDK inhibitors (eg. Flavopiridol and SNS-032) are not selective to CDK9, inhibiting other CDKs and enzymes. Their lack of selectivity and decreased potency may contribute to many adverse effects in clinical trials [8]. Therefore, selective inhibitors of CDK9 are needed to prevent the undesirable off-target effects and to enhance potency.

AZD 4573 is highly potent against CDK9 ($<3$ nM $IC_{50}$) and selective ($>10$ fold) against CDK9. The drug results in caspase activation and loss of viability across a diverse set of hematological cancers including MM [16]. MC180295 is also a highly selective CDK9 inhibitor ($>22$ fold, $IC_{50} = 5$ nM) that has broad anti-cancer activity in vitro and in vivo [17]. In this study, we noted that AZD 4573 and MC180295 *in vitro* inhibited the viability of MM cells. We also showed that AZD 4573 is synergistic with ARV 825 in inducing apoptosis and inhibiting MM cell proliferation both *in vitro* and *in vivo*.

## Materials and methods

### Cell culture

Human MM cell lines: KMS11, KMS28, KMS18, KMS12, MM1S, MM1R, H929, 8226, 8226 LR5 and 8226 P100V were kind gifts from Dr. W.J. Chng (Cancer Science Institute of Singapore, Singapore) and KMS11 res and MM1S res were generous gifts from Dr. A.K. Stewart (Mayo Clinic, Arizona). 293 FT cells was obtained from Prof. Koeffler's lab (Cancer Science Institute of Singapore, Singapore), it was cultured and maintained in DMEM with 10% fetal bovine serum (FBS). STR analysis was done on all cell lines used in this study. All other cell lines were cultured and maintained in RPMI1640 with 10% FBS and 1% penicillin-streptomycin (Invitrogen, Carlsbad, CA) at 37°C with 5% $CO_2$. The 8226 LR5 cells were maintained in 10 nM Melphalan, the 8226 P100V cells were cultured with 100 nM bortezomib for 2 days every 2 weeks.

### Cell proliferation assay

Twenty thousand cells were seeded in 96-well plates followed by drug treatment. After 72 h culture, 10 μl of MTT (2-(4,5-dimethylthiazol-2-yl)-2,5-diphenyltetrazolium bromide) was added to the wells and cultured at 37°C for an additional 4 h followed by addition of 100 μl stop solution (10% Sodium Dodecyl Sulphate). Plates were measured with a spectrophotometer at 570 nm absorbance. $IC_{50}$ values were calculated using Graph Pad Prism.

## Annexin V and propidium iodide (Annexin V-PI) apoptosis analysis

Cells were treated with different concentrations of ARV 825 for 48 h. Staining was performed using Apoptosis Detection Kit II (BD Biosciences, USA). Cells were harvested and washed twice with phosphate-buffered saline (PBS, Life technologies, USA), suspended in 1X binding buffer with 5 μl of FITC conjugated Annexin V and 5 μl of PI for 15 min in the dark at room temperature. Samples were analyzed using flow cytometric analysis (Sony SA3800). Cells positive for Annexin and PI were defined as apoptotic cells.

## Cell cycle analysis

Cells were treated with different concentrations of AZD 4573 (24 h), fixed with 70% chilled ethanol, washed with PBS two times and stained with PI solution [40 μg/ml PI, Triton X-100 (1%), 20 ug/ml DNase-free RNase A in PBS] for 30 min at 37˚C in the dark followed by flow cytometric analysis (Sony SA3800).

## Drug combination study

Results from MTT assays with different combinations of ARV 825 and AZD 4573 were evaluated by CompuSyn [18] (ComboSyn, Inc, Paramus, NJ). A combination index (CI) plot is a Fa-CI plot in which CI<1, = 1, >1 indicate synergism, additive and antagonism, respectively. Fa: fraction of proliferation inhibition by the drug.

## Reagents and antibodies

ARV 825 was developed by the C.M. Crew's laboratory (Department of Chemistry, Yale University, New Haven, CT, USA). We obtained the drug from Chemietek (Indianapolis, IN, USA), Catalog no. CT-ARV825. MC180295 was a generous gift from Dr. H.H. Zhang and J.P. Issa (Temple University, Philadelphia). We obtained AZD 4573 from MedChemExpress (New Jersey, USA), Catalog no. HY-112088. For *in vitro* administration, ARV 825, AZD 4573 and MC180295 were dissolved in dimethyl sulfoxide (Sigma-Aldrich) (10 mM) and stored at -80˚C. List of antibodies and inhibitors is present in S1 Table.

## Western blot analysis

Cellular lysates were prepared using M-PER mammalian protein extraction reagent (Thermo Scientific, Rockford, USA) containing 1X protease cocktail inhibitor (Roche, Switzerland). After 20 min incubation on ice, lysates were centrifuged (14,000g, 30 min, 4˚C). Total protein concentrations were measured by Pierce Coomassie Plus (Bradford) assay kit (Thermo Fisher Scientific). Twenty micrograms of protein were loaded per lane on SDS-PAGE gel and resolved at 90 voltages, followed by transfer to PVDF (Millipore, Massachusetts). Membranes were blocked with 5% non-fat milk and incubated with antibodies.

## Lentiviral production and silencing of CDK9

shRNA targeting CDK9 was cloned into pLKO.1 lentiviral vector (Sequence: Forward: `CCGG GTTCGACTTCTGCGAGCATGACTCGAGTCATGCTCGCAGAAGTCGAACTTTTG`Reverse:`AAT TCAAAAAGTTCGACTTCTGCGAGCATGACTCGAGTCATGCTCGCAGAAGTCGAC`. Luciferase vector was purchased from Addgene (plasmid #17477). Recombinant lentiviral vector and packaging vector (pCMV-dR8.9 and pMD2.G-VSVG) were cotransfected into 293 FT cells using polyethylenimine (PEI) according to the manufacturer's instructions. Virus supernatants were harvested at 48h and 72h after transfection, and placed through a 0.45 μm filter. 8226 and KMS28 cells (1 X $10^6$ per well) were seeded in 6-well plates. Cells were transduced with

lentiviral vectors in the presence of 8 μg/ml polybrene (Sigma-Aldrich) for 24 h. Stable cell lines were selected with puromycin.

## In vivo xenografts

*In vivo* studies were performed with a protocol approved by the Institutional Animal Care and Use Committee at Cedars Sinai Medical Center. The mice were purchased from Charles River Laboratories. The mice were kept at Animal Research Facilities at Cedars Sinai Medical Center in a sterile condition with proper food, water and environment. They were monitored daily by trained comparative medicine staffs for the health and well-being. To access the *in vivo* activity of ARV 825, KMS11 expressing luciferase (KMS11$^{LUC}$) were injected into lateral tail vein of SCID-Beige mice. Mice were monitored for 7 days and imaged by Xenogen IVIS spectrum (PerkinElmer, Massachusetts) camera to document engraftment before treatment was initiated. At 7 days after mice were injected with cells, they were randomly divided into four groups (5 mice in each group) [vehicle (5% Kolliphor® HS15), 5 mg/kg of ARV 825 (intraperitoneal injection daily for 28 days), 10 mg/kg of AZD 4573 (intraperitoneal injection, twice a day with 2 h interval for two consecutive days/week for 4 weeks) and combination of ARV 825 and AZD 4573]. Tumor burden in each treatment group was monitored daily and imaged weekly by Xenogen camera for 28 days. The mice were then euthanized within 24 hours after the end of experiment. Mice health and behavior was monitored daily. No mice died or paralyzed before the end of experiment. For euthanasia, the mice received isoflurane overdose for anesthesia followed by cervical dislocation. All research personnel in mice study were trained for animal care and welfare according to IACUC protocol approved by Cedars Sinai Medical Center.

## Statistical analysis

For *in vitro* and *in vivo* experiments, the statistical significance of difference between two groups used two-tailed student t-test and two-way ANOVA. GraphPad Prism 6 software (GraphPad Software Inc., San Diego, CA, USA) was used for all calculations. Data were presented as means ± SD. Asterisks in the figures represent significant differences between experimental groups in comparison to controls (* $p < 0.01$, ** $p < 0.001$, *** $p < 0.0001$). Data points in figures represent means ± SD (standard deviation).

## Results

### CDK9 inhibitors (AZD 4573 and MC180295) decreased cellular proliferation of MM cells

AZD 4573 and MC180295 (CDK9 inhibitors) in a dose-dependent manner were tested against a panel of 12 human MM cell lines (KMS11, MM1R, KMS12BM, H929, KMS18, 8226 LR5, MM1S, KMS11 res, 8226, KMS28, 8226 P100V, MM1S res) using an in vitro proliferation assay (MTT, 72 h). Cell lines included melphalan-resistant (8226 LR5), steroid-resistant (MM1R), bortezomib-resistant (8226 P100V) and lenalidomide-resistant (KMS11res and MM1Sres) cell lines. Some of the cell lines have cytogenetics associated with a poor prognosis [e.g. t(4:14): KMS11, KMS28, H929; t(14:16): MM1S, 8226]. All MM cell lines were sensitive to AZD 4573 with IC$_{50}$ ranging from 8–60 nM (Fig 1A). MM1S, MM1Sres, MM1R and KMS11 cell lines (IC$_{50}$ = 8 nM) were most sensitive to AZD 4573; whereas 8226 P100V was a relatively more resistant cell line (IC$_{50}$ = 70 nM) (S2 Table). The data showed that AZD 4573 was effective even if the cells were resistant to either melphalan, lenalidomide, steroid, bortezomib or they had a cytogenetically unfavorable chromosome. MC180295 was not as potent as AZD

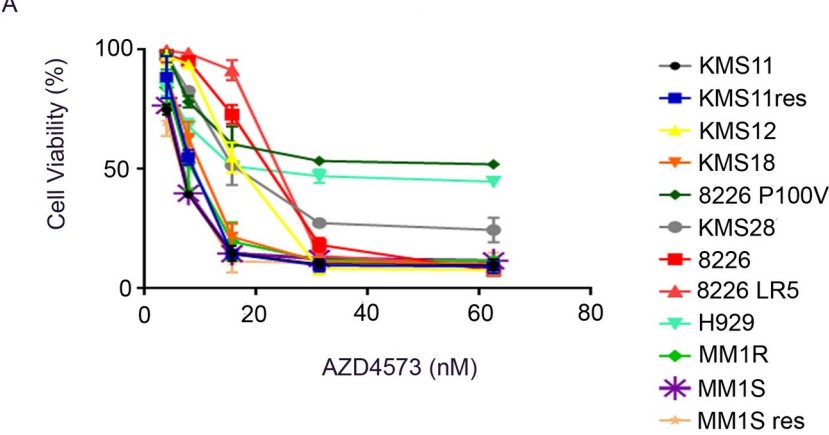

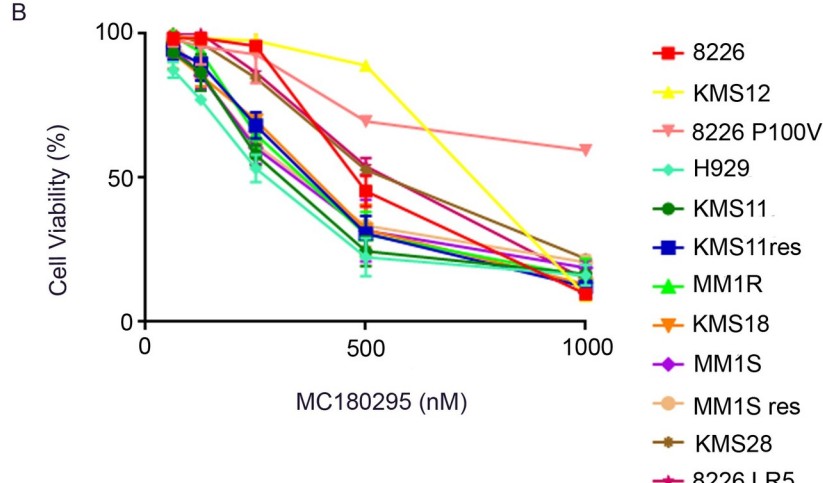

**Fig 1. CDK9 inhibitors: Anti-proliferative activities against MM cells.** (A) Twelve MM cell lines were cultured with AZD 4573 (1 nM-62.5 nM, 72 h). Growth inhibition was measured by MTT assays. Results are mean ± SD, N = 3. $IC_{50}$s are shown in S2 Table. (B) MM cells treated with MC180295 (1 nM-1,000 nM, 72 h). Growth inhibition was measured by MTT assays. Results are mean ± SD, N = 3. $IC_{50}$s are shown in S2 Table.

4573 against MM cell lines with IC50 ranging from 260 nM to >1000 nM (Fig 1B). H929 was the most sensitive ($IC_{50}$ = 260 nM), whereas 8226 P100V was the relatively resistant cell line to MC180295 ($IC_{50}$ > 1000 nM) (S2 Table).

## Silencing of CDK9 reduced cell proliferation and viability of MM cells

To examine the dependency of MM cells on CDK9 expression, we performed short hairpin RNA (shRNA)-mediated silencing of CDK9 in KMS28 and 8226 cell lines. RT-qPCR and western blot analysis confirmed the successful silencing of CDK9 using shRNA in these cells (Fig 2A and 2B, left upper and lower panel). Silencing CDK9 in 8226 and KMS28 cell lines reduced cell viability compared to control cells (MTT, 72 h) (Fig 2A and 2B, right panel).

A

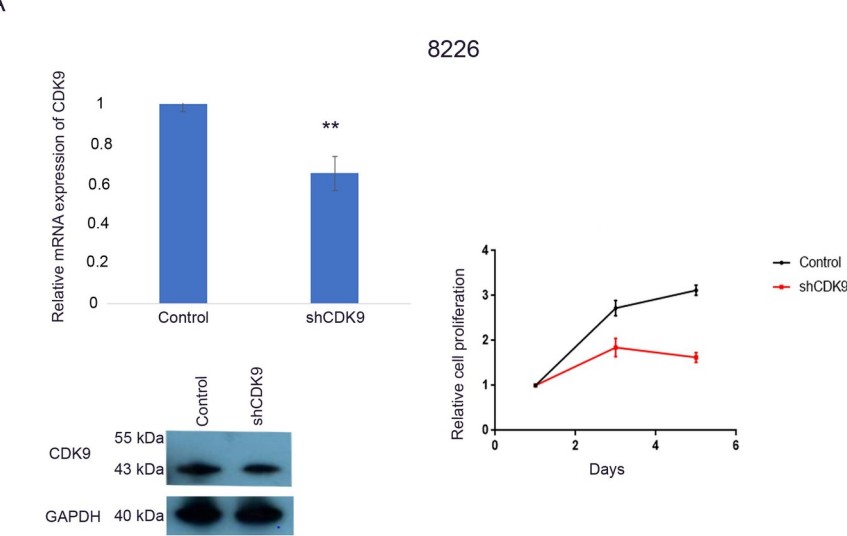

B

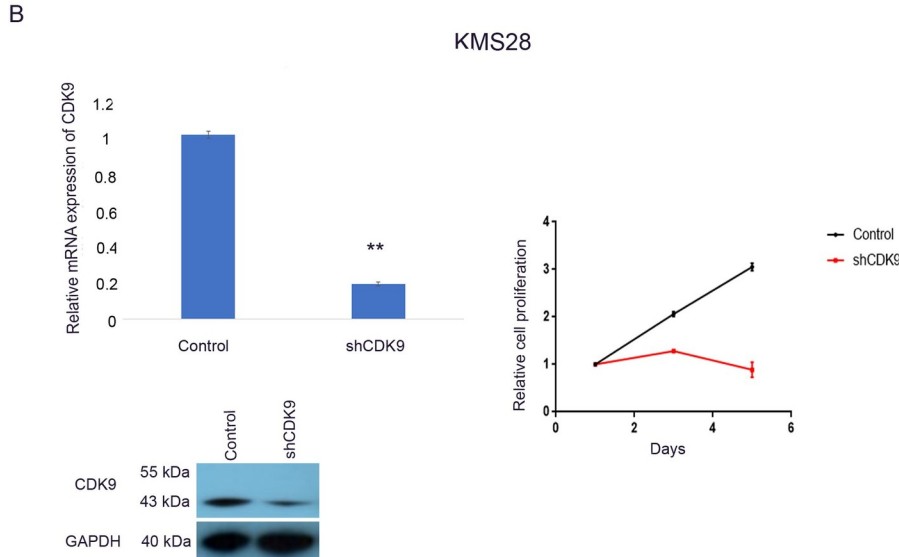

**Fig 2. shRNA mediated silencing of CDK9 decreased proliferation of MM cells.** (A) Levels of CDK9 mRNA (Left, upper panel) and protein (Left, lower panel) after shRNA-mediated silencing of 8226 cells. Cell proliferation assay (MTT) after shRNA silencing of 8226 cells (right panel). (B) Levels of CDK9 mRNA (left, upper panel) and protein (left, lower panel) after shRNA-mediated silencing of CDK9 in KMS28 cell line. Cell proliferation assays (MTT) after shRNA silencing of CDK9 in KMS28 cell line (right panel). Results are mean ± SD, N = 3.

## ARV 825 and AZD 4573 showed synergistic growth inhibitory activity against MM cell lines

The IC$_{50}$ of ARV 825 against KMS11, 8226 and KMS28 MM cells are 9 nM, 84 nM and 137 nM, respectively (S3 Table). Combination of ARV 825 and AZD 4573 treatment for 72h

showed synergistic growth inhibitory activity against these MM cells (Combination Index < 1) (Fig 3). The combination index analysis is shown in S4 Table.

## CDK9 inhibitor AZD 4573 downregulated phosphorylation of ser2 pol II CTD, MCL-1 and MYC protein

We evaluated the protein expression of BRD 2, BRD 3, BRD 4, phosphorylated ser2 pol II carboxy terminal domain (Pol II CTD), total RNA polymerase II, MCL-1 and MYC in KMS11 and 8226 cells after treatment with ARV 825 and AZD 4573 either as a single agent or in combination (7 h, ARV 825 [KMS11 (20 nM; 40 nM); 8226 (100 nM; 200 nM)], AZD 4573 [KMS11 (20 nM; 40 nM); 8226 (60 nM; 120 nM)].

Protein levels of phosphorylated RNA pol II, MCL-1 and MYC decreased significantly after KMS11 and 8226 cells were treated with AZD 4573 [KMS11 (20 nM; 40 nM); 8226 (60 nM; 120 nM, 7 h) whereas the total RNA pol II was not affected. In contrast, after ARV 825 treatment [KMS11 (20 nM;40 nM); 8226 (100 nM; 200 nM), 7 h] of KMS11 and 8226 cells, protein expression of BRD 2, BRD 3, BRD 4 and MYC reduced, but did not affect the protein expression of MCL-1 and phosphorylated RNA pol II (Fig 4A and 4B). Combination treatment (7 h) of ARV 825 [KMS11 (20 nM;40 nM); 8226 (100 nM; 200 nM)] and AZD 4573 [KMS11 (20 nM; 40 nM); 8226 (60 nM; 120 nM)] of KMS11 and 8226 cells markedly reduced their protein expression of BRD 2, BRD 4, MYC and phosphorylated RNA pol II as compare to single agents alone (Fig 4A and 4B).

## AZD 4573 and ARV 825 combination markedly induced apoptosis in MM cells

Flow cytometric analysis of KMS11 and 8226 MM cells showed a marked increase in the percentage of apoptotic cells after treatment with combination of ARV 825 and AZD 4573 for 48 h compare to control cells. Either ARV 825 (2.5 nM) or AZD 4573 (2.5 nM) alone produced 26% and 13% apoptotic KMS11 cells, respectively; but their combination (2.5 nM + 2.5 nM) led to 67% apoptotic cells. Similarly, 8226 cells treated with either ARV 825 (20 nM) or AZD 4573 (10 nM) alone led to 14% and 20% of apoptotic cells, respectively; but their combination (20 nM + 10 nM) led to 71% of apoptotic cells (Fig 5A).

Cell cycle analysis of MM cells was performed in the presence of various concentrations of AZD 4573 for 24 h compare to control cells. AZD 4573 only produced a minimal dose-dependent increase in G1 phase, decrease S phase and G2/M phase in KMS11 and 8226 cell lines (Fig 5B).

Combination Index: AZD4573 + ARV825, synergistic growth inhibition

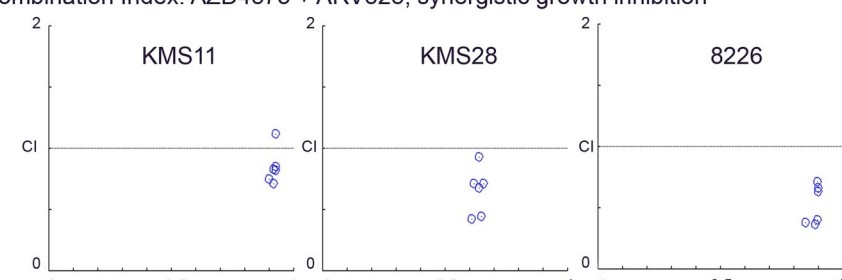

**Fig 3. Combination index plot of ARV 825 with AZD 4573 against MM cells.** Synergistic growth inhibition of KMS11, KMS28 and 8226 cells when ARV 825 and AZD 4573 are combined (72 h, MTT assay). CI < 1, CI = 1 and CI > 1 represent synergism, additive, and antagonism respectively of the combination of the two compounds. Values of Combination Index analysis are shown in S4 Table.

A

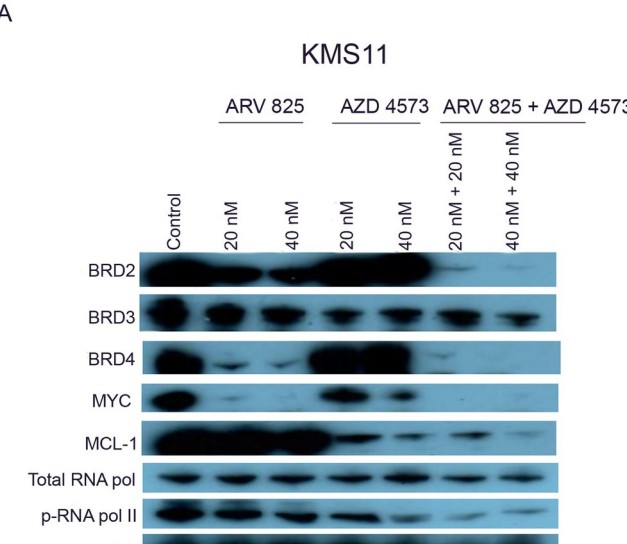

B

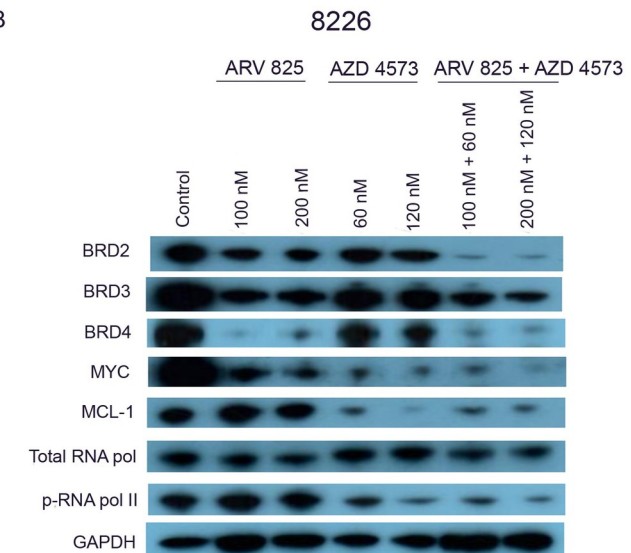

**Fig 4. Effect of ARV 825 and AZD 4573 on protein expression of BRD 2, BRD 3, BRD 4; phosphorylated Ser 2 RNA pol II; total RNA pol II; MCL-1 and MYC in MM cells.** (A) KMS11 cells were treated with ARV 825 (20 nM and 40 nM), AZD 4573 (20 nM and 40 nM) and their combination [ARV 825 + AZD 4573 (20 nM + 20 nM; 40 nM + 40 nM, respectively) for 7 h; and protein expression was examined by western blot (GAPDH, internal control). (B) 8226 cells were treated with ARV 825 (100 nM and 200 nM), AZD 4573 (60 nM and 120 nM) and their combination [ARV 825 + AZD 4573 (100 nM + 60 nM; 200 nM + 120 nM, respectively) for 7 h; and protein expression was examined by western blot.

## Combination of AZD 4573 and ARV 825 inhibited MM cells in vivo

Anti-proliferative effect of either ARV 825 or AZD 4573 alone or their combination was examined *in vivo* against MM xenografts growing in SCID Beige mice. One week after injection, the MM cells were observed by bioluminescence imaging; after which, mice (n = 5 per group) were randomly assigned to receive ARV 825 (5 mg/kg, IP daily), AZD 4573 (10 mg/kg, IP,

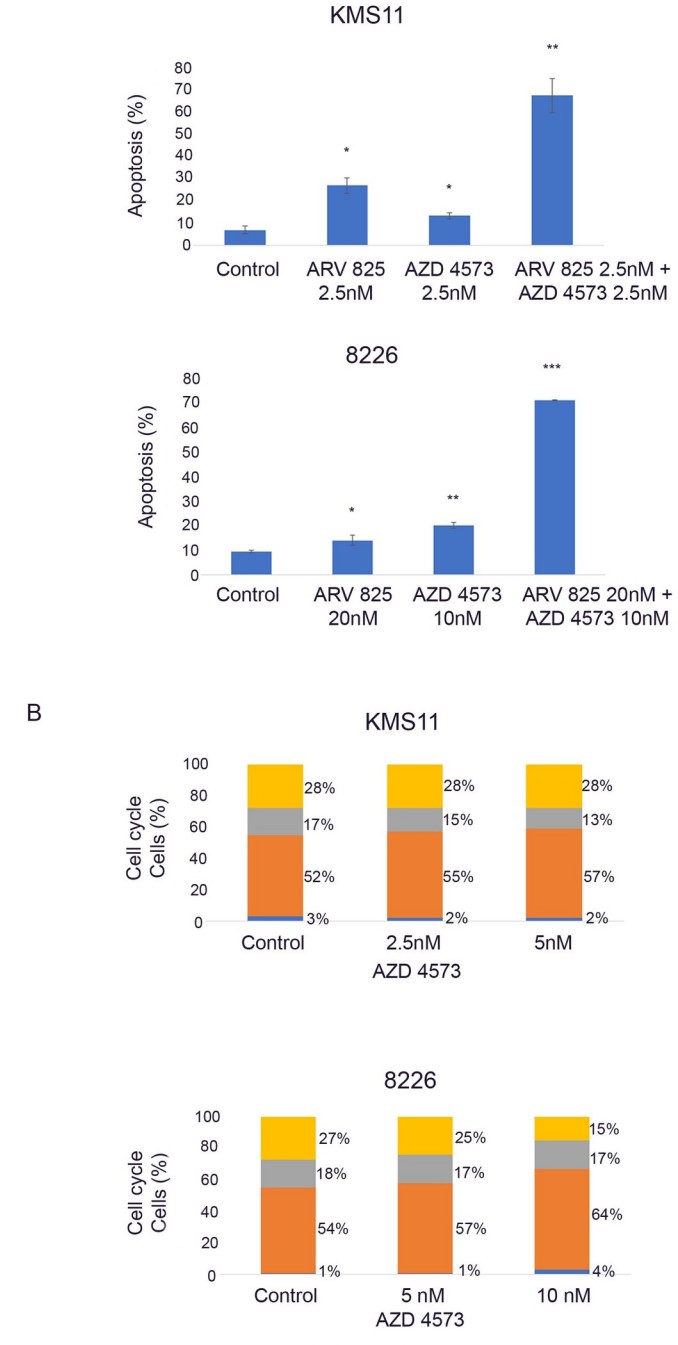

**Fig 5. Apoptosis and cell cycle analysis of MM cells after treatment with ARV 825 and/or AZD 4573.** (A) Apoptosis: KMS11 and 8226 MM cells were treated with of ARV 825 (2.5 nM; 20 nM), AZD 4573 (2.5 nM; 10 nM) and their combination (2.5 nM + 2.5 nM; 20 nM + 10 nM, respectively) for 48 h, stained with annexin V-FITC and PI, and analyzed by flow cytometry. Histograms represent percentage of apoptotic cells. Mean ± SD of three independent experiments. (B) Cell cycle: KMS11 and 8226 MM cells were treated for 24 h with either AZD 4573 (2.5–5 nM or 5–10 nM, 24 h), respectively or diluent control (DMSO), stained with propidium iodide (PI) and analyzed by flow cytometry. Histograms showed proportion of cells in different phases of cell cycle. Representative of three independent experiments. $^*p \leq 0.01$; $^{**}p \leq 0.001$; $^{***}p \leq 0.0001$ for ARV 825 vs. control.

A

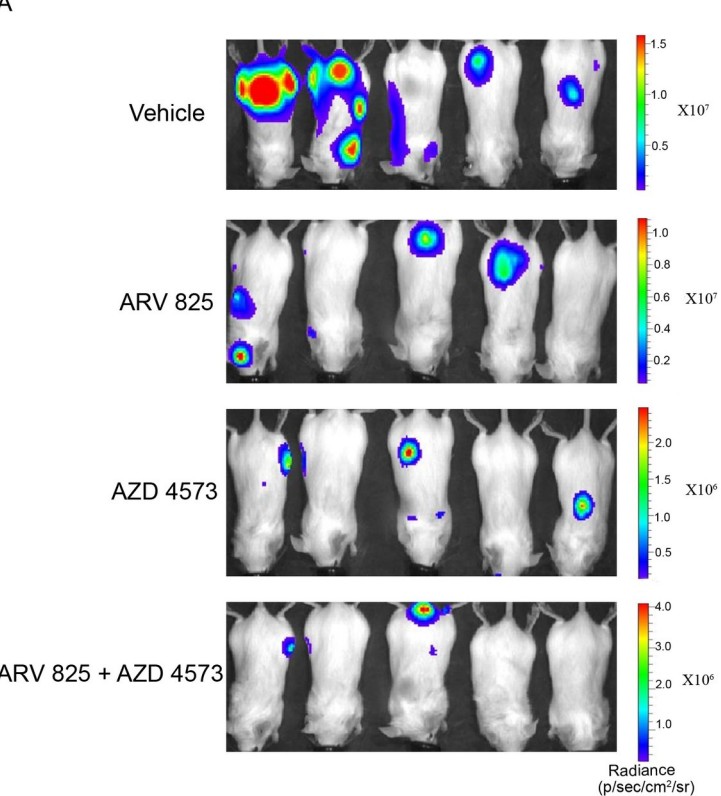

B

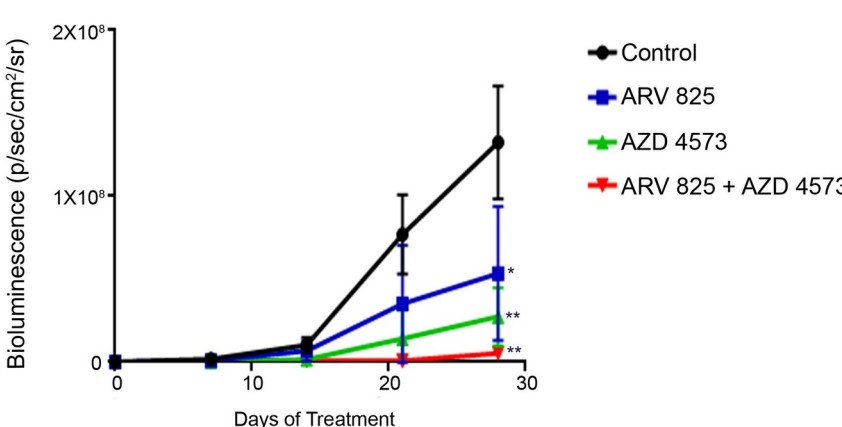

**Fig 6. AZD 4573 acts synergistically with ARV 825 in inhibiting MM cells in vivo.** (A) Whole-body bioluminescence images of SCID-beige mice after intravenous injection with KMS11[LUC] cells followed 7 days later by treatment with either ARV 825 (5 mg/kg IP daily for 28 days) or AZD 4573 (10 mg/kg, IP, twice a day with 2 h interval for two consecutive days/week for 4 weeks) as well as a combination of both drugs and vehicle control alone. (B) Tumor burden as measured by bioluminescence in SCID-beige mice after intravenous injection with KMS11[LUC] cells. Data represent mean ± SD (N = 5 per group). $^*p \leq 0.01$, $^{**}p \leq 0.001$, $^{***}p \leq 0.0001$.

twice a day with 2 h interval for two consecutive days/week) or combination treatment for a total duration of 28 days. Control mice received vehicle alone. Bioluminescence was measured at days 0, 7, 14, 21, 28. Combination of ARV 825 and AZD 4573 significantly (P < 0.001) slowed tumor growth in experimental mice compared to single agent alone as measured by bioluminescence (Fig 6A and 6B) at days 21 and 28. ARV 825 and AZD 4573 alone or in combination did not affect either the normal activity or the weight (loss < 10%) of the mice (S1 Fig).

## Discussion

Management of multiple myeloma remains challenging especially relapse/refractory MM despite major advancement in treatment. Therefore, new targeted therapies are urgently required. BET PROTAC and CDK9 inhibitors have shown promising results in preclinical studies against MM [4,5,19,20]. However, the limited clinical efficiency of CDK9 inhibitors due to side-effects and dose-limiting toxicities have prevented these drugs from receiving FDA approval. Hence, the need for better CDK9 inhibitors [8]. In addition, combination therapies targeting multiple survival pathways may minimize adverse effects by reducing dosage and improving outcomes. Furthermore, targeting BRD 4 when also targeting CDK9 is able to block the compensatory increase in expression of MYC [15].

AZD 4573 is a selective CDK9 inhibitor that led to dose- and time- dependent decrease in phosphorylated ser 2 RNAP II and loss of MCL1 mRNA and protein; also, the in vivo efficacy of the drug has been reported in multiple hematological tumors [16]. It is currently in phase I clinical trials for treatment of hematological malignancies [21]. MC180295 is a novel CDK9 inhibitor that is more selective against CDK9 than AZD 4573. However, MC180295 was not as potent as AZD 4573 against MM cell lines. AZD4573 markedly decreased growth of almost all the MM cells having resistance to standard drugs. We performed shRNA mediated silencing of CDK9 against MM cells and found it paralleled the drugs' activity to inhibit proliferation and viability of MM cells.

In this study, we also demonstrated that combination of the BET PROTAC ARV 825 and the selective CDK9 inhibitor AZD 4573 synergistically caused significant growth inhibition of myeloma cells both in vitro and in an orthotopic xenograft model. Also, flow cytometric analysis demonstrated that low dose combination of ARV 825 and AZD 4573, as compare to single agent, induced enhanced apoptosis. These lower drug concentrations will minimize unwanted off target activity or side-effects. We observed only minimal G1 cell cycle arrest in KMS11 and 8226 cells after treatment with AZD 4573 suggesting that AZD 4573 probably does not inhibit the cell cycle as a mechanism of cell kill but it does block transcriptional elongation [8].

We found that in MM cells AZD 4573 decreased phosphorylation of RNAP II, decreased anti-apoptotic proteins (MCL-1 and MYC), whereas ARV 825 degraded BET proteins and decreased expression of MYC. Combining both inhibitors synergistically inhibited cell growth of MM cells. Prior studies showed that inhibition of CDK9 paradoxically increased expression of MYC [15]. This did not occur with our drug combination. Our therapeutic targeting of MYC is important in MM because of the importance of this protein causing progression of MM. In summary, our studies showed for the first time that the combination of a BET PROTAC (ARV 825) plus AZD 4573 (CDK9 inhibitor) is effective against MM.

## Supporting information

**S1 Fig. Mice weight after treatment.** Comparison of weight of mice after treatment with ARV 825 (5 mg/kg IP daily for 28 days), AZD 4573 (10 mg/kg, IP, twice a day with 2 h interval for two consecutive days/week for 4 weeks), combination of both drugs or diluent control.

Mean ± SD of 5 mice in each group.
(TIF)

**S1 Raw images.**
(PDF)

**S1 Table. List of antibodies.**
(DOCX)

**S2 Table. IC50s of AZD 4573 against MM cells, 72 h.**
(DOCX)

**S3 Table. IC50s of ARV 825 against MM cells, 72 h.**
(DOCX)

**S4 Table. Combination index.** Combination index of AZD 4573 synergistic with ARV 825
(CI < 1, CI = 1 and CI > 1 represent synergism, additive and antagonism, respectively).
(DOCX)

## Acknowledgments

We express gratitude to Jean-Pierre Issa and Hang-Hang Zhang for providing the MC180295
inhibitor.

## Author Contributions

**Conceptualization:** Liang Xu, Bing-Chen Han, Pavithra Shyamsunder.

**Data curation:** Su-Lin Lim.

**Formal analysis:** Su-Lin Lim.

**Funding acquisition:** H. Phillip Koeffler.

**Investigation:** Liang Xu, Bing-Chen Han, Pavithra Shyamsunder, H. Phillip Koeffler.

**Methodology:** Su-Lin Lim, Liang Xu, Bing-Chen Han, Pavithra Shyamsunder.

**Resources:** Pavithra Shyamsunder, Wee-Joo Chng.

**Supervision:** Wee-Joo Chng, H. Phillip Koeffler.

**Writing – original draft:** Su-Lin Lim.

**Writing – review & editing:** Su-Lin Lim, Liang Xu, Bing-Chen Han, Pavithra Shyamsunder,
Wee-Joo Chng, H. Phillip Koeffler.

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
