## [Decision Letter · Decision Letter 0]

8 May 2020

PONE-D-20-08881

Multiple Myeloma: Combination Therapy of BET Proteolysis Targeting Chimeric Molecule with CDK9 Inhibitor

PLOS ONE

Dear Dr Lim,

Thank you for submitting your manuscript to PLOS ONE. After careful consideration, we feel that it has merit but does not fully meet PLOS ONE’s publication criteria as it currently stands. Therefore, we invite you to submit a revised version of the manuscript that addresses the points raised during the review process. In particular, as noted by the reviewer below, some of the figures need revision for quality and presentation.  

We would appreciate receiving your revised manuscript by Jun 22 2020 11:59PM. To enhance the reproducibility of your results, we recommend that if applicable you deposit your laboratory protocols in protocols.io, where a protocol can be assigned its own identifier (DOI) such that it can be cited independently in the future. For instructions see: http://journals.plos.org/plosone/s/submission-guidelines#loc-laboratory-protocols

We look forward to receiving your revised manuscript.

Kind regards,

Arun Rishi, Ph.D.

Academic Editor

PLOS ONE

Journal Requirements:

2. At this time, we request that you  please report additional details in your Methods section regarding animal care, as per our editorial guidelines:

(1) Please state the source of them mice used in the study  

(2) Please provide details of animal welfare (e.g., shelter, food, water, environmental enrichment)

(3) Please describe any steps taken to minimize animal suffering and distress, such as by administering anaesthesia  

(4) Please describe the post-operative care received by the animals, including the frequency of monitoring and the criteria used to assess animal health and well-being.

Thank you for your attention to these requests.

3. Please provide additional information about each of the cell lines used in this work, including any quality control testing procedures (authentication, characterisation, and mycoplasma testing). For more information, please see http://journals.plos.org/plosone/s/submission-guidelines#loc-cell-lines.

In addition, please provide the source, culture conditions and quality control testing procedures of the HEK 293 FT cells used in your study.

4. In the Methods section, please provide the product number and any lot numbers of the ARV-825 and AZD 4573 purchased from chemical companies for your study.

5. To comply with PLOS ONE submission guidelines, in your Methods section, please provide additional information regarding your statistical analyses, including the software used for the analysis. For more information on PLOS ONE's expectations for statistical reporting, please see https://journals.plos.org/plosone/s/submission-guidelines.#loc-statistical-reporting.

Reviewers' comments:

Reviewer's Responses to Questions

**Comments to the Author**

1. Is the manuscript technically sound, and do the data support the conclusions?

Reviewer #1: Yes

2. Has the statistical analysis been performed appropriately and rigorously? 

Reviewer #1: Yes

3. Have the authors made all data underlying the findings in their manuscript fully available?

Reviewer #1: Yes

4. Is the manuscript presented in an intelligible fashion and written in standard English?

Reviewer #1: Yes

5. Review Comments to the Author

Reviewer #1: Multiple Myeloma: Combination therapy of BET Proteolysis Targeting Chimeric Molecule with CDK9 Inhibitor.

The author reported for the first time that the combination of a BET PROTAC(ARV 825) plus CDK9 inhibitor(AZD 4573) is effictive for the treatment of muitiple myeloma. It is worthy of publication in PLOS ONE after major revision of the following points.

1. Figure 1 to 6 are not clear enough;

2. Western Blot Analysis needs to be redone.

6. PLOS authors have the option to publish the peer review history of their article (what does this mean?). If published, this will include your full peer review and any attached files.

Reviewer #1: No

---

## [Author Response · Author response to Decision Letter 0]

22 May 2020

Author’s reply:

Thank you for the comment. The revised manuscript is corrected to meet the PLOS ONE’s style requirements.

2. At this time, we request that you please report additional details in your Methods section regarding animal care, as per our editorial guidelines:

(1) Please state the source of the mice used in the study 

Author’s reply: 

The source of the mice is now included in revised manuscript in Methods section. Line 161-162

(2) Please provide details of animal welfare (e.g., shelter, food, water, environmental enrichment)

Author’s reply: 

It is included in revised manuscript in Methods section. Line 161-164

(3) Please describe any steps taken to minimize animal suffering and distress, such as by administering anaesthesia 

Author’s reply: 

As suggested, it has been included in revised manuscript in Methods section. Line 174-176

(4) Please describe the post-operative care received by the animals, including the frequency of monitoring and the criteria used to assess animal health and well-being.

Author’s reply: 

The animal health and well-being were monitored daily by trained staffs at comparative medicine according to the approved IACUC protocol by Cedars Sinai Medical Center.

3. Please provide additional information about each of the cell lines used in this work, including any quality control testing procedures (authentication, characterisation, and mycoplasma testing). For more information, please see http://journals.plos.org/plosone/s/submission-guidelines#loc-cell-lines

Author’s reply: 

As recommended, the additional information about cell lines used is included in the revised manuscript in Materials and Methods section. Line 89-92. STR analysis was done on all cell lines.

In addition, please provide the source, culture conditions and quality control testing procedures of the HEK 293 FT cells used in your study. 

Author’s reply:

The information about 293FT cells is now included in revised manuscript in Materials and Methods section. Line 92-94.

4. In the Methods section, please provide the product number and any lot numbers of the ARV-825 and AZD 4573 purchased from chemical companies for your study.

Author’s reply:

As suggested, the catalog number for both ARV 825 and AZD 4573 are included in revised manuscript in Methods section. Line 130 and 132.

5. To comply with PLOS ONE submission guidelines, in your Methods section, please provide additional information regarding your statistical analyses, including the software used for the analysis. For more information on PLOS ONE's expectations for statistical reporting, please see https://journals.plos.org/plosone/s/submission-guidelines.#loc-statistical-reporting.

Author’s reply:

The information regarding statistical analysis is included in revised manuscript in Methods section. Line 180-182.

Reviewer’s comment:

1. Is the manuscript technically sound, and do the data support the conclusions?

Reviewer #1: Yes

Author’s reply: Thank you for the comment

2. Has the statistical analysis been performed appropriately and rigorously?

Reviewer #1: Yes

 Author’s reply: Thank you for the comment

3. Have the authors made all data underlying the findings in their manuscript fully available?

Reviewer #1: Yes

Author’s reply: Thank you for the comment

4. Is the manuscript presented in an intelligible fashion and written in standard English?

Reviewer #1: Yes

Author’s reply: Thank you for the comment

5. Review Comments to the Author

Reviewer #1: Multiple Myeloma: Combination therapy of BET Proteolysis Targeting Chimeric Molecule with CDK9 Inhibitor.

The author reported for the first time that the combination of a BET PROTAC(ARV 825) plus CDK9 inhibitor(AZD 4573) is effective for the treatment of multiple myeloma. It is worthy of publication in PLOS ONE after major revision of the following points.

1. Figure 1 to 6 are not clear enough;

2. Western Blot Analysis needs to be redone.

Author’s reply: Thank you for the comment. The quality of figures has been improved as suggested in revised manuscript

---

## [Decision Letter · Decision Letter 1]

8 Jun 2020

Multiple Myeloma: Combination Therapy of BET Proteolysis Targeting Chimeric Molecule with CDK9 Inhibitor

PONE-D-20-08881R1

Dear Dr. Lim,

We’re pleased to inform you that your manuscript has been judged scientifically suitable for publication and will be formally accepted for publication once it meets all outstanding technical requirements.

Kind regards,

Arun Rishi, Ph.D.

Academic Editor

PLOS ONE

Additional Editor Comments (optional):

Reviewers' comments:

Reviewer's Responses to Questions

**Comments to the Author**

1. If the authors have adequately addressed your comments raised in a previous round of review and you feel that this manuscript is now acceptable for publication, you may indicate that here to bypass the “Comments to the Author” section, enter your conflict of interest statement in the “Confidential to Editor” section, and submit your "Accept" recommendation.

Reviewer #1: (No Response)

2. Is the manuscript technically sound, and do the data support the conclusions?

Reviewer #1: (No Response)

3. Has the statistical analysis been performed appropriately and rigorously? 

Reviewer #1: (No Response)

4. Have the authors made all data underlying the findings in their manuscript fully available?

Reviewer #1: (No Response)

5. Is the manuscript presented in an intelligible fashion and written in standard English?

Reviewer #1: (No Response)

6. Review Comments to the Author

Reviewer #1: (No Response)

7. PLOS authors have the option to publish the peer review history of their article (what does this mean?). If published, this will include your full peer review and any attached files.

Reviewer #1: No

---

## [Editor Report · Acceptance letter]

10 Jun 2020

PONE-D-20-08881R1 

Multiple Myeloma: Combination Therapy of BET Proteolysis Targeting Chimeric Molecule with CDK9 Inhibitor 

Dear Dr. Lim:

I'm pleased to inform you that your manuscript has been deemed suitable for publication in PLOS ONE. Congratulations! Your manuscript is now with our production department. 

Kind regards, 

on behalf of

Prof Arun Rishi 

Academic Editor

PLOS ONE